# Human Uncertainty-Aware Reliable Data Selection and Efficient Annotation for Visual Question Answering

## Abstract

Large vision-language models (VLMs) achieve strong performance but still depend on supervised fine-tuning (SFT) with massive annotated datasets, which are both costly and inherently noisy due to human annotation, especially when human uncertainty exists. We find that the degree of human uncertainty affects the reliability of a sample, thereby casting doubt on its suitability for SFT. It is still unknown how to use human uncertainty for training when such imperfect data exists. Moreover, current mainstream SFT method simply requires annotation for the full dataset, causing unnecessary annotation overhead. In this work, we revisit Visual Question Answering (VQA), one of the most important and commonly studied task for VLMs. We study data reliability and label efficiency based on VQA. To this end, we propose a **h**uman **u**ncertainty-aware **r**eliable data selection and efficient label **a**nnotation method (HURA). HURA's advantages are twofold: firstly, it filters harmful samples and prioritizes more reliable samples that indeed improving model performances (both accuracy and human alignment), reducing computational costs. Secondly, it does not require huge amount of human annotation on overall dataset, reducing human annotation costs and avoiding potential manual noise. We find that training with only a small random subset (~10%) of data already recovers most of the full-data performance (~90%), while not all samples are equally reliable to improve model performances: high human uncertainty samples contribute little or even do harm to training, while medium- and low- human uncertainty samples provide more improvements. We also find that models are able to proceed self-training with a provided seed set, thereby reducing both annotation reliance and cost. Our experimental results demonstrated that HURA is effective for recent state-of-the-art VQA models on VQAv2 dataset. HURA highlights an important direction for learning reliably from imperfect data: understanding and leveraging uncertainty, rather than simply scaling up the size of training data. Future code and data link will be here.[1]

## 1 Introduction

Visual Question Answering (VQA) [5] is a widely studied task for evaluating large vision-language models (VLMs) [3, 14, 18, 8, 11, 12]. Its task formulation—requiring models to jointly reason over visual content and natural language questions, making it a strong indicator of vision-language understanding and generation capabilities.

---

[1]Our code and data are well prepared and will be public available after anonymous period.

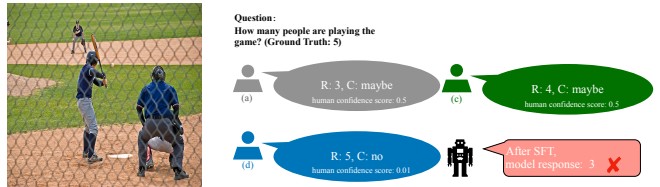

Figure 1: Different human annotators have different answers without clear certainty. 'R' indicates human response while 'C' is the confidence annotation. The ground truth is added as an indicator only in the Figure but it is not available in the dataset or training. For samples with mixed annotations, models may not learn correct answers and yield wrong predictions.

However, despite the performances of current state-of-the-art (SOTA) models [3, 14, 19], we observe that current models suffer from several sources of unreliability, such as high human uncertainty in training data, fine-tuning strategies, and the evaluation metric. Firstly, as illustrated in Figure 1, we observe many VQA samples exhibit substantial human uncertainty, where multiple annotators provide different yet low-confidence answers. Such uncertainty casts doubt on whether these samples should be treated as equally effective training data. Two previous work has illustrated the problem of not all samples are suitable for VQA training, but neither of them studied large VLMs or human uncertainty [17, 9]. Secondly, fine-tuning methods for VLMs on VQA predominantly rely on simple supervised fine-tuning (SFT) [4, 16]. While straightforward, this approach requires massive human annotation, significantly increasing both labeling and computational costs. More importantly, such "full dataset" training may fail to fully exploit the value of individual samples: prioritizing more informative examples can potentially enable models to achieve comparable or even superior performance with fewer data. Thirdly, evaluation metric VQA-Accuracy [1] commonly used in VQA research also exhibit unreliability. The dominant VQA accuracy metric, simply and only based on frequency matching of annotator responses, is prone to misrepresenting model quality, especially on high-uncertainty samples. Recent work [11] has shown that SOTA VLMs struggle to align with human confidence levels , which challenges their reliability, as a reliable model should 'know what it knows' [2, 15, 6]. However, Lan et al. [11] has not proposed new training methods to address the problems revealed. Developing more reliable VQA models still remains a challenge.

In this work, we adopt the recently proposed human uncertainty-aware evaluation metric HUD [11], replacing the traditional VQA accuracy with HUD score for a more accurate evaluation. Unlike frequency-based metrics, HUD explicitly accounts for human disagreement and uncertainty in annotations, enabling a more comprehensive and faithful evaluation. Based on HUD, we evaluate how samples with different uncertainty levels influence SFT. Following prior work [11, 17], we categorize samples into three levels of uncertainty: high, medium, and low. Our analysis reveals that medium- and low-uncertainty samples are far more beneficial for training, whereas high-uncertainty samples contribute little and may even degrade performance. To the best of our knowledge, this is the first work to reveal the differential impact of human uncertainty levels on VQA training, which we believe offers valuable guidance for future research on data-efficient multi-modal learning. Additionally, our results show that training on only (~10%) of data achieves (~90%) of the full-data performance. Our insight is that a more efficient and reliable training pipeline is needed. Therefore, we propose HURA (Human Uncertainty-aware Reliable data selection and efficient Annotation), a method that filters harmful samples, prioritizes reliable ones, and enables self-training from a small seed set. We validate HURA on the widely used VQA dataset, VQAv2 [5], and demonstrate HURA's effectiveness on different SOTA models.

Our main contributions are as follows: 1) To the best of our knowledge, we are the first work to reveal how samples with different levels of human uncertainty affect VQA SFT. 2) We propose HURA, a novel data selection and training strategy, with practical gains in efficiency and reliability: HURA achieves competitive performance while reducing annotation and computational costs. HURA's confidence distribution also aligns better with human uncertainty distribution, indicating HURA helps in achieving more reliable VLMs.

## 2 Background

**Human Uncertainty and HUD score in VQA.** Given an image-question-answer triplet $t = (i, q, \mathcal{A})$, where $i$ is an image, $q$ is a question, and $\mathcal{A}$ is an answer set, $\mathcal{A}$ consists of 10 independent humans' annotations. In each annotation $h_n$, $n = 1, \ldots, 10$, every annotator gives response $r_n$ and confidence level $c_n$. The level $c_n$ belongs to one of the three pre-defined categories $< \text{'yes'}, \text{'no'}, \text{'maybe'} >$, indicating whether an annotator is confident in their answer. To quantify human uncertainty, HUD assigns different human confidence scores, quantifying every 'yes' as 0.99, 'no' as 0.01, and 'maybe' as 0.5 respectively. The HUD score is then determined by averaging these human confidence scores across responses to the same question, and a human distribution $H = [h_1, h_2, ..., h_m]$ is quantified, where $m$ is the number of different responses. High HUD scores indicate low human uncertainty. In this work, we map samples scores between [0,0.33) as high uncertainty, [0.33,0.66) as medium uncertainty, and [0.66, 0.99] as low uncertainty, evenly dividing the overall distribution into three parts. Due to the page limit, more details can be found in the original work [12].

**Model Confidence Distribution and Evaluation Methods.** Given a sample $x$, the prediction distribution of a model $\mathcal{M}$, denoted as $\mathcal{M}(x)$, corresponds to the probabilities $P_Y(Y = y | X = x)$ the model assigns to each class $y$. Given a human distribution $H = [h_1, h_2, ..., h_m]$, the model probability distribution over multiple classes is denoted as: $M_Y = softmax([l_1, l_2, ..., l_m])$, where $l_i$ is the logits of a label in the last hidden layer. Traditionally, the model is evaluated over VQA-AccuracyZ [1], where $Acc(\text{ans}) = \min\left\{\frac{\#\text{humans that said ans}}{3}, 1\right\}$. It is maximized (1.0), if at least 3 raters gave the exact answer. The number 3 is a manually fixed parameter in the original work [1]. Previous work also use **KL-divergence** (**KL** Kullback and Leibler [10]) to measure how the model probability distribution diverges from human's, denoted as KL(Human||Model).

## 3 HURA pipeline

The pipeline of HURA is presented in Algorithm 1. Let $S$ denote a small *seed set* with human annotations, $M$ a pre-trained VLM without SFT, and $R$ a large *remaining dataset* without annotations. HURA aims to fine-tune from $M$ to $M_T$ by pseudo-labeling samples from $R$. HURA first removes high human-uncertainty samples from $S$, obtaining a filtered subset $S_{\text{filt}}$. The base model $M$ is fine-tuned on $S_{\text{filt}}$ to obtain an updated model $M_1$. On the held-out portion of $S$, we compute KL divergences between $M_1$'s predictions and human distributions for three uncertainty levels (high, medium, low), the mean value denoted as $K_1, K_2, K_3$. We also compute a *threshold distribution $\tau$*, defined as the mean human distribution over low-uncertainty samples. We iterate over samples $r \in R$ in small batches (e.g., $1\%$ of $R$ at a time). For each $r$, we use the current model $M_i$ to compute the predictive distribution $p_{M_i}(r)$ and its KL divergence to $\tau$. If this KL divergence lies in the interval $[K_1, K_2]$, the sample is retained and assigned a pseudo-label $\hat{y}_r = \arg\max p_{M_i}(r)$; otherwise, it is discarded. The retained samples are then used to fine-tune $M_i$ to $M_{i+1}$. This process repeats until a predefined number of iterations $T$ is reached.

## 4 Experiments and Results

**baselines and evaluation metrics.** We evaluate three open-sourced recent SOTA VLMs: LLaVA-1.5-7B [13], Qwen2-VL-7B [18], and BEiT3 [19]. Unlike previous works, we do not use VQA-Accuracy due to the mentioned shortcomings in this paper, but replace the frequency with the HUD score of each different response, we denote this evaluation as HUD-acc. We evaluate on the validation set because the test set is not public available. The reason to start only on VQAv2 is similar to previous work [11], but we expect to expand to other dataset like VizWiz [7] in the future.

**Results and Discussion.** In Table 1, we find that even though simply training on the overall set still achieves the best performances, HURA reaches very similar performances on all three models, with only 20 percent of training data. This indicates that HURA successfully reduces the reliance on both annotation and computation. Moreover, HURA reaches a more reliable confidence distribution with humans with lower KL scores. This indicates that simply training on large amount of data may ruin model's confidence level, while HURA aligns better with humans.

It is also clearly observed that for BEiT3, simply training on randomly selected ~10% samples results in ~90% performances compared with training on all set. On models like LLaVA and Qwen which

---

**Algorithm 1:** HURA: Human Uncertainty-aware Reliable Data Selection and Self-Training

**Input:** Seed set $S$; pre-trained VLM $M$ (no SFT); unlabeled pool $R$; max iterations $T$.
**Output:** Fine-tuned model $M_T$.

1 **Step 1: Seed fine-tuning with uncertainty filtering**;
2 $S_{\text{filt}} \leftarrow S \setminus S_{\text{high}}$;               `// remove high human-uncertainty samples`
3 Fine-tune $M$ on $S_{\text{filt}}$ to obtain $M_1$;
4 **Step 2: Calculating KL thresholds**;
5 Compute $K_1, K_2, K_3$ as the (mean) KL divergences between $M_1$ predictions and human distributions on {high, medium, low} subsets from the held-out part of $S$;
6 Compute threshold distribution $\tau$ as the mean human distribution on the low-uncertainty subset;
7 **Step 3: Iterative selection & self-training**;
8 **for** $i \leftarrow 1$ **to** $T$ **do**
9     Sample a mini-batch $B \subset R$ (e.g., 1% of $R$);
10     $S_{\text{sel}} \leftarrow \emptyset$;
11     **foreach** $r \in B$ **do**
12        $p \leftarrow p_{M_i}(r)$;               `// model predictive distribution on r`
13        $d \leftarrow \text{KL}(p \,\|\, \tau)$;
14        **if** $K_1 \leq d \leq K_2$ **then**
15           $\hat{y}_r \leftarrow \arg\max p$;
16           $S_{\text{sel}} \leftarrow S_{\text{sel}} \cup \{(r, \hat{y}_r)\}$;
17     Fine-tune $M_i$ on $S_{\text{sel}}$ to obtain $M_{i+1}$;

---

| Model | Set split or method | HUD-Acc ↑ | KL(H$\|$M) ↓ |
|-------|---------------------|-----------|--------------|
| BEiT3 | All (SFT) | **72.41** | 1.8221 |
|       | 10% Low+Med (SFT) | 65.25 | 1.6496 |
|       | High (SFT) | 2.74 | - |
|       | HURA (ours, T=10) | **72.81** | **1.3269** |
| LLaVA-1.5 | zero-shot | **73.01** | 0.6953 |
|           | All (SFT) | **76.41** | 0.6010 |
|           | 10% Low+Med (SFT) | 74.73 | 0.5235 |
|           | High (SFT) | 72.50 | 0.6863 |
|           | HURA (ours, T=10) | **76.38** | **0.4973** |
| Qwen2VL | zero-shot | **74.01** | 0.6879 |
|         | All (SFT) | **76.15** | 0.5612 |
|         | 10% Low+Med (SFT) | 75.51 | 0.5331 |
|         | High (SFT) | 73.73 | 0.6674 |
|         | HURA (ours, T=10) | **76.08** | **0.4903** |

Table 1: Model performances on HUD-Accuracy (best results and HURA are highlighted in bold.) and KL divergence (best results are highlighted in bold. Due to space constraints and more experiments are still ongoing, we only present and discuss the results in Table 1 in this submission.

has zero-shot capability, the performances are even higher with limited data. On the contrary, it is also shown that high human uncertainty samples do harm to training, where it reaches lower accuracy compared with the zero-shot performances. Particularly, on BEiT 3, since it does not have zero-shot ability, simply training on high uncertainty samples totally make the training fails.

## 5   Conclusion, limitation, and future work

In this work, we revisit VQA and proposed a method HURA to study how use human uncertainty information more efficiently and also more accurately so that we can reach a more reliable model. We evaluate HURA on VQAv2 and demonstrate its effectiveness from both accuracy and confidence distribution perspective. We will not limit to this mainstream dataset but expand to more datasets such as VizWiz in the future. For the more ongoing experiments, we are expecting to find out exact reasons why high-human uncertainty fails to help training, and how the other remaining samples work to help model. We also expect to come up with more experimental results indicating the how different turns of iterations of HURA helps training. These are already ongoing experiments and we expect to expand them to become a solid future work. Due to the 4-page limitation, more details of clarity (e.g., the method part) will also be expanded in the future publication.

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
