# OpenReview forum: "Human Uncertainty-Aware Reliable Data Selection and Efficient Annotation for Visual Question Answering"
_NeurIPS.cc/2025/Workshop/Reliable_ML — NeurIPS 2025 - Reliable ML Workshop_

### Official Review · Reviewer_of2z · 2025-09-12
**Reasonable but Incomplete and Unclear**

**Rating:** 6
**Confidence:** 4

**Review:**

The idea of incorporating confidence levels into the SFT process is reasonable, and the authors’ approach to implementing this seems sound. However, the paper leaves important aspects unclear -- in particular, the proportion of low-confidence data instances is not specified (e.g., what if very few images elicit low-confidence responses?). Moreover, the absence of results and comprehensive experiments makes the work feel incomplete. Overall, while the concept is promising, the paper appears to be a work in progress, and I look forward to seeing a more complete version.

---

### Official Review · Reviewer_zrEx · 2025-09-20
**Promising work, but deeper analysis needed to strengthen the claims**

**Rating:** 7
**Confidence:** 2

**Review:**

# *Summary*

The paper approaches data selection from the angle of human label variation and confidence as a data selection attribute.  The method takes a subset of training samples, filters out high-uncertainty samples, fine-tunes only on medium- and low-uncertainty data, and then uses a KL-based self-training pipeline to pseudo annotate the rest of the training data, keeping only instances where HLV matches. Results on VQAv2 show that HURA can achieve near full-data performance using only \~20% of annotated data.

# *Strengths.*

- Attempts to integrate human uncertainty into both training and evaluation.
- Empirical evidence that full-data accuracy is recoverable from a reduced subset.

# *Weaknesses / Limitations.*

- Related work on human label variation could be more broad.
- Distribution of high uncertainty data in the test split is unknown since the test set is not public. But to fairly evaluate the proposed method, we do need to know if it is the case that models generally don’t learn from high uncertainty samples, even when included in training or does the test set not contain a good representation of high uncertainty samples?


*Subjective Weakness:* The paper sometimes seem to treat human label variation as inherently negative, using terms like “harmful samples” and “manual noise.” However, there is substantial prior work arguing the opposite, in that human label variation can provide useful signal (e.g., [The "Problem" of Human Label Variation: On Ground Truth in Data, Modeling and Evaluation](https://aclanthology.org/2022.emnlp-main.731/), [Git-repol awesome-human-label-variation](https://github.com/mainlp/awesome-human-label-variation)). I therefore disagree with the paper’s framing in that regard.

# *Suggestions for Authors*

* Consider constructing a holdout evaluation set stratified by uncertainty levels, and test HURA on it. This would make clear whether the method truly generalizes across uncertainty bins.
* Consider reporting VQA-Acc alongside HUD metrics.
* Why do the authors hypothesise that BEiT3 performs so poorly on high uncertainty data compared to the other two models?